# Assembly and Functional Role of PACE Transporter PA2880 from *Pseudomonas aeruginosa*

Jiangfeng Zhao,[a,b] Nils Hellwig,[c] Bardya Djahanschiri,[d] Radhika Khera,[b] Nina Morgner,[c] Ingo Ebersberger,[d,e,f] Jingkang Wang,[a] Hartmut Michel[b]

[a]Tianjin University, School of Chemical Engineering and Technology, State Key Laboratory for Chemical Engineering, Collaborative Innovation Center of Chemical Science and Chemical Engineering, Tianjin, People's Republic of China

[b]Department of Molecular Membrane Biology, Max Planck Institute of Biophysics, Frankfurt am Main, Germany

[c]Institute of Physical and Theoretical Chemistry, Goethe University Frankfurt, Frankfurt am Main, Germany

[d]Department for Applied Bioinformatics, Institute for Cell Biology and Neuroscience, Goethe-University Frankfurt, Frankfurt am Main, Germany

[e]Senckenberg Biodiversity and Climate Research Centre Frankfurt (BIK-F), Frankfurt am Main, Germany

[f]LOEWE Centre for Translational Biodiversity Genomics, Frankfurt am Main, Germany

**ABSTRACT** The recently identified proteobacterial antimicrobial compound efflux (PACE) transporters are multidrug transporters energized by the electrochemical gradient of protons. Here, we present the results of phylogenetic and functional studies on the PACE family transporter PA2880 from *Pseudomonas aeruginosa*. A phylogenetic analysis of the PACE family revealed that PA2880 and AceI from *Acinetobacter baumannii* are classified into evolutionarily distinct clades, although they both transport chlorhexidine. We demonstrate that PA2880 mainly exists as a dimer in solution, which is independent of pH, and its dimeric state is essential for its proper function. Electrogenicity studies revealed that the chlorhexidine/H$^+$ antiport process is electrogenic. The function of several highly conserved residues was investigated. These findings provide further insights into the functional features of PACE family transporters, facilitating studies on their transport mechanisms.

**IMPORTANCE** *Pseudomonas aeruginosa* is a pathogen that causes hospital-acquired (nosocomial) infections, such as ventilator-associated pneumonia and sepsis syndromes. Chlorhexidine diacetate is a disinfectant used for bacterial control in various environments potentially harboring *P. aeruginosa*. Therefore, investigation of the mechanism of the efflux of chlorhexidine mediated by PA2880, a PACE family transporter from *P. aeruginosa*, is of significance to combat bacterial infections. This study improves our understanding of the transport mechanism of PACE family transporters and will facilitate the effective utilization of chlorhexidine for *P. aeruginosa* control.

**KEYWORDS** *Pseudomonas aeruginosa*, PACE family, PA2880, dimer, electrogenic process

Resistance against antimicrobials has become one of the greatest health problems in recent years (1). The multidrug resistance (MDR) transporters play key roles in this process by promoting the increase of multidrug-resistant strains of many bacterial pathogens (2–4). Several studies indicate that secondary metabolites and antibacterial toxins might be the naturally occurring substrates of MDR transporters (5, 6). Among the currently known MDR transporters (7–10), the proteobacterial antimicrobial compound efflux (PACE) family transporters, with AceI from *Acinetobacter baumannii* as the prototype, are the most recently identified bacterial drug efflux transporters (11). Despite low sequence similarities between the small multidrug resistance (SMR) family transporters and the PACE family transporters, the transporters of the two families possess similar sizes and secondary structures. They can actively transport structurally diverse antimicrobial compounds, including benzalkonium, dequalinium, acriflavine, proflavine and chlorhexidine. In addition, it has been reported that

Address correspondence to Jingkang Wang, jkwang@tju.edu.cn, or Hartmut Michel, Hartmut.Michel@biophys.mpg.de.

The authors declare no conflict of interest.

short-chain diamines, which play vital roles in metabolism, transcription regulation, and protein expression (12, 13), might be the physiological substrates of PACE family transporters (14).

As secondary active transporters, PACE transporters utilize the electrochemical proton gradient across the membrane as the energy source for the active export of their substrates (14, 15). Members of the PACE family are mainly present in proteobacteria and have so far not been found in archaea and eukaryotes (10). On the basis of amino acid sequence similarities and functional studies, the PACE transporters are classified into two clades: the chlorhexidine-responsive clade and the chlorhexidine-unresponsive clade (15). A glutamic acid residue located in the predicted transmembrane helix 1 (TM1) is fully conserved in both groups, and it has been suggested that this residue is essential for proton binding (14, 15). In a recent study, the functional unit of AceI was reported to be a dimer and the conserved glutamic acid (residue E15 in AceI) was shown to be vital for its dimerization (16). In addition, the study also showed that the dimeric and monomeric forms of AceI exist in a dynamic equilibrium in solution and that the equilibrium is dependent on the pH. The dimer population dominates at high pH, and protonation of residue E15 at low pH results in an increase of the monomeric form (16). Similarly, the SMR family transporters also function as dimers and the glutamic acid residue located in TM1 is suggested to be involved in both proton and substrate binding at different stages of the translocation cycle (17). A more comprehensive study of the PACE family is still missing.

In this work, we determined the phylogenetic profiles of two PACE family transporters encoded in the genome of *A. baumannii*, AceI and A1S_1503, across the proteobacterial diversity, with a particular focus on the human pathogen *Pseudomonas aeruginosa*. A maximum-likelihood (ML) phylogenetic tree reconstruction revealed a highly reticulated tree in the case of AceI, which indicates that the corresponding protein in *P. aeruginosa* strain PAO1, PA2880, belongs to a different subfamily within the PACE family. Functional analysis showed that PA2880 is a proton-dependent chlorhexidine transporter. Five highly conserved residues (E38, D83, H101, E106, and D132) that might be responsible for the cation and substrate binding were identified. The results of electrogenicity studies show that the chlorhexidine/H$^+$ antiport process is electrogenic. Moreover, we demonstrate that PA2880 mainly exists as a dimer in solution, which is independent of pH, and that the dimerization of PA2880 is essential to its proper function.

## RESULTS AND DISCUSSION

**Phylogenetic analysis of the PACE family.** PACE family transporters play an essential role in the evolutionary emergence of bacterial resistances against antimicrobial substances. Thus far, studies examining the function of these transporters have focused mainly on AceI from *Acinetobacter baumannii*. To see to what extent insights from studying AceI can be generalized to other members of the PACE family, in particular to those encoded in the genome of the human pathogen *Pseudomonas aeruginosa*, we determined the phylogenetic profile of AceI across 1,364 strains representing 1,209 species spanning the entire diversity of proteobacteria. We could identify 399 orthologs in 383 taxa (Table S1 in the supplemental material), among them PA2880 from *Pseudomonas aeruginosa* PAO1. A maximum-likelihood phylogeny reveals that the evolutionary relationships of these sequences are complex and do not reflect the relationships of the taxa they are found in (Fig. 1, Fig. S1). Most importantly, we find that the AceI orthologs of *A. baumannii* and of *P. aeruginosa*, which belong to the same bacterial order, *Pseudomonadales*, are placed in two well-separated clades (ML bootstrap support 91), where each clade harbors sequences from a different bacterial phylum (*Betaproteobacteria*). Moreover, the *Acinetobacter* AceI orthologs are only distantly related to any other sampled member of this subfamily. Their patristic distance, i.e., the sum of the branch lengths in the phylogeny, to the closest related sequence outside the genus is 2.5, more than twice the distance between PA2880 and a sequence from the betaproteobacterium *Polaromonas naphthalenivorans* (*Burkholderiales*) (Fig. S1). Given the isolated placement of the *Acinetobacter* AceI orthologs (supplemental text and Fig. S2), we conclude that the two transporters, AceI and PA2880, have diverged to such an extent that substantial differences in their mode of action can be expected.

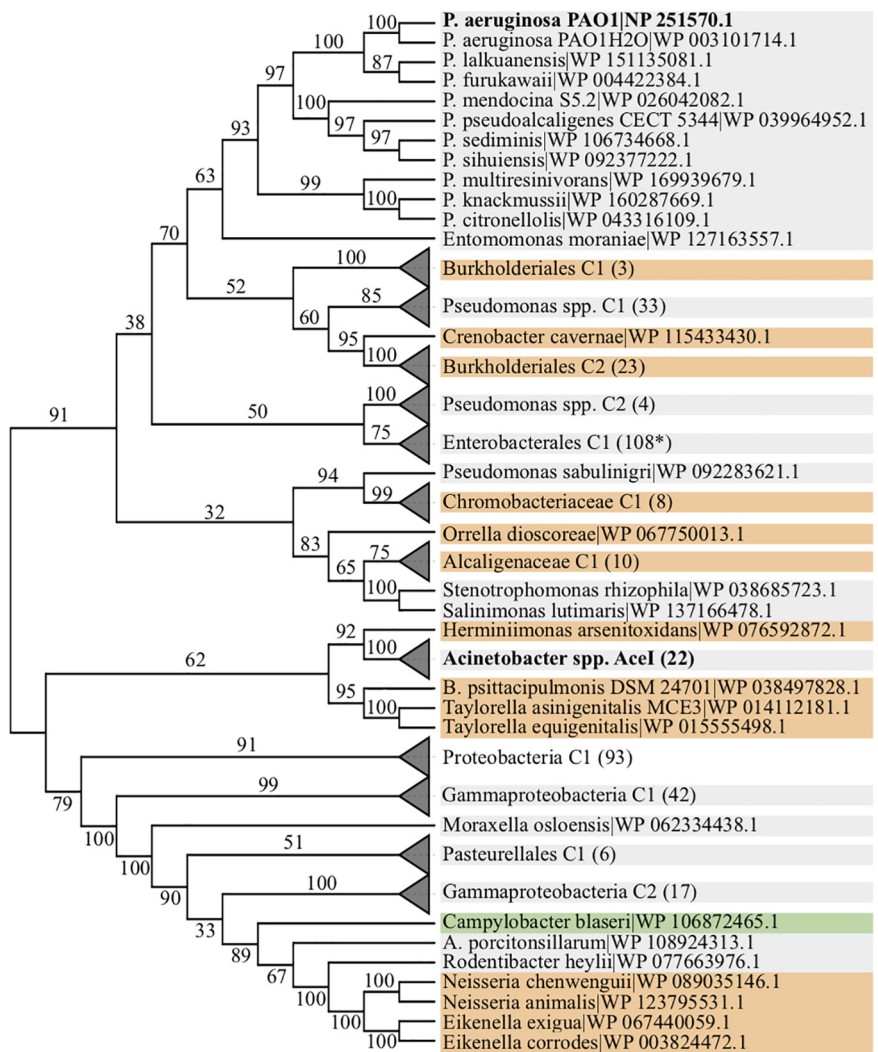

**FIG 1** Maximum-likelihood phylogeny of 399 *A. baumannii* AceI orthologs. Taxa are grouped to higher-order systematic groups where applicable to facilitate a better overview. The two focal clades, *Acinetobacter* spp. and *P. aeruginosa*, are shown in bold face. Branch labels indicate percent bootstrap support, and branch lengths are not drawn to scale. Node labels indicate the species or, alternatively, the higher-order systematic group together with the number of species subsumed in the group. RefSeq protein accession numbers are given next to the species name. The background of the node labels indicates the bacterial class, as follows: gray, *Gammaproteobacteria*; orange, *Betaproteobacteria*; green, *Epsilonproteobacteria*; white, *Proteobacteria*. The fully expanded tree, together with branch length information, is shown in Fig. S1.

**Expression of PA2880 in *E. coli* leads to a higher cellular resistance to chlorhexidine but not cadaverine.** A previous study on *Pseudomonas aeruginosa* indicated that PA2880 was increasingly expressed when the bacterium was subjected to chlorhexidine exposure (18). To confirm the role of PA2880 in the export of chlorhexidine, an antimicrobial susceptibility test was performed using *E. coli* BL21 cells, because this strain does not possess the genes that encode endogenous PACE family proteins. The results show that the expression of PA2880 fused with enhanced green fluorescent protein (PA2880-eGFP) in BL21 cells led to a higher tolerance toward chlorhexidine than was seen in control cells (Table 1). To exclude the possible effect of eGFP, the same experiment was performed using BL21 cells expressing PA2880. The results indicated a lower cellular resistance to chlorhexidine upon PA2880 expression, as indicated by decreased MICs (Table S2). This observation might be caused by the lower expression level of PA2880 as opposed to PA2880-eGFP. Earlier studies reported that transporters from the resistance/nodulation/division (RND) superfamily, the small multidrug resistance (SMR) family, and the multidrug and major

**TABLE 1** Drug resistance levels of *E. coli* BL21 cells expressing PA2880-eGFP and its variants

| Protein expressed | MIC of[a]: | |
| --- | --- | --- |
| | Chlorhexidine ($\mu$g/mL) | Cadaverine (mg/mL) |
| Control | 0.5 | 1.25 |
| Wild type | 2 | 1.25 |
| E38Q | 0.5 | — |
| W74A | 1 | — |
| D83A | 1 | — |
| H101A | 1 | — |
| E106A | 0.5 | — |
| D132A | 0.5 | — |

[a]—, MIC value was not determined. Experiments were repeated at least 3 times.

facilitator superfamily (MFS) could also mediate the efflux of chlorhexidine (19–21). To minimize the effect of multidrug transporters present in the *E. coli* BL21 cells, an antimicrobial susceptibility test was carried out using the *E. coli* drug-hypersensitive strain BW25113 ($\Delta acrB$ $\Delta emrE$ $\Delta mdfA$) (22). A lower MIC than that determined with *E. coli* BL21 cells was obtained, which is compatible with chlorhexidine export by these multidrug pumps, but the presence of PA2880 still increased the cellular resistance to chlorhexidine (Table S2). Altogether, PA2880 mediates resistance to chlorhexidine. On the other hand, our results show that PA2880 is incapable of causing resistance of *E. coli* BL21 cells to cadaverine (Table 1), which was previously reported to be a native substrate of the PACE family transporter AceI (14). These results raise questions about the physiological role of PA2880 and encourage further investigations.

A previous study identified four amino acid sequence motifs in PACE family proteins that are close to the cytoplasmic boundaries of the transmembrane helices (TMs) (23). The sequence motifs in TM1 (motif 1A) and TM3 (motif 1B) feature a membrane-embedded glutamic residue (the sole residue [E15 in the case of AceI, referred to as E15$^{AceI}$] that has been functionally characterized until now) (14–16) and histidine and arginine residues at the membrane boundary. The motifs found in TM2 (motif 2A) and TM4 (motif 2B) notably contain several aromatic residues and aspartate residues embedded in the membrane bilayer. To explore the functional importance of these residues, six highly conserved residues, including E38$^{PA2880}$ in motif 1A, W74$^{PA2880}$ and D83$^{PA2880}$ in motif 2A, H101$^{PA2880}$ and E106$^{PA2880}$ in motif 1B, and D132$^{PA2880}$ in motif 2B, were selected based on the sequence alignment results (Fig. S3). *In vivo* drug susceptibility studies demonstrated that residue E38 in PA2880 (corresponding to E15 in AceI) is essential for chlorhexidine resistance, which is consistent with previous reports (10, 15). The replacement of residues E106 and D132 by alanine residues totally abolished transport activity, and in the W74A, D83A, and H101A variants, the resistance to chlorhexidine was decreased to a smaller extent (Table 1). A Western blot analysis showed that all variants were expressed at similar levels as wild-type PA2880 (Fig. S4). Except for W74A, the variants also maintained their structural integrity, as indicated by the gel filtration and blue native-polyacrylamide gel electrophoresis (BN-PAGE) (Fig. S5 to S7). Therefore, the W74A variant was not investigated further.

**Efflux of chlorhexidine mediated by PA2880 is driven by proton motive force.** The PACE family transporters are reported to employ the electrochemical gradient of protons to drive the substrate efflux (14). To examine whether the coupled ions are protons in the case of PA2880, chlorhexidine transport mediated by PA2880 was performed in a reconstituted system, and pH changes inside the lumen of proteoliposomes were monitored using the membrane-impermeable, pH-sensitive fluorescent dye 8-hydroxyprene-1,3,6-trisulfonic acid (pyranine). PA2880 was purified and reconstituted into liposomes composed of *E. coli* polar lipids. Empty liposomes and proteoliposomes containing TqsA (an autoinducer 2 transporter, which was used as a negative control) (24) were prepared in the same manner for comparison. When an electrochemical gradient across the membrane (acidic and positive inside) was applied, the subsequent addition of 100 $\mu$M chlorhexidine to PA2880 proteoliposomes

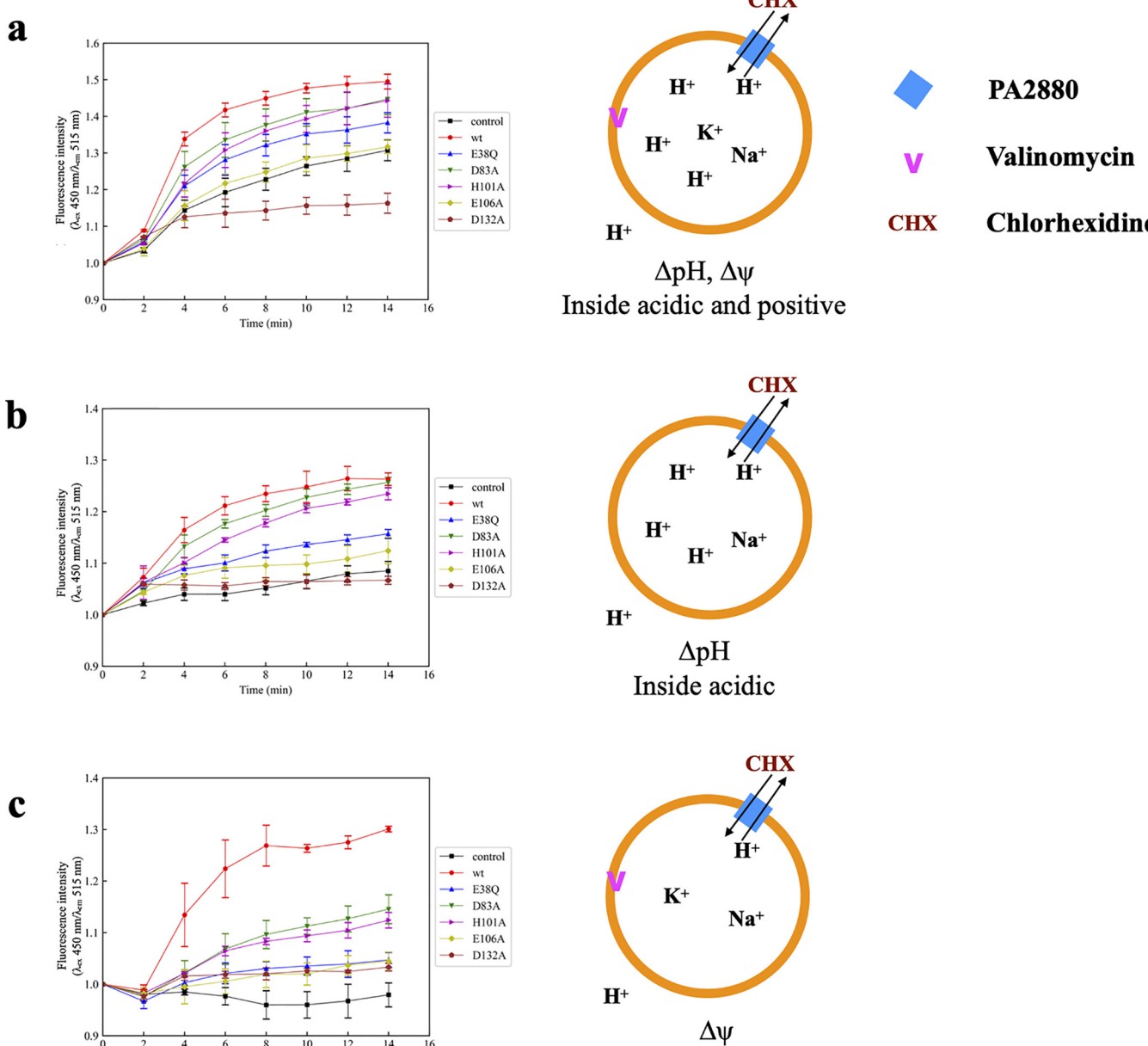

**FIG 2** *In vitro* transport activity of PA2880 and its variants. Proteoliposomes were prepared in buffer containing the pH-sensitive fluorescent dye pyranine (see Materials and Methods). The excitation maximum of pyranine shifted from 400 nm to 450 nm with increasing pH, while the emission maximum was stable at ~515 nm. Therefore, the fluorescence intensity measured at 515 nm with an excitation wavelength of 450 nm was used to monitor the pH change in the lumen of proteoliposomes, and an increase of this value is indicative of the increase of internal pH. All (proteo-)liposomes were prepared in Na$^+$ buffer at pH 7. (a) (Proteo-)liposomes were diluted into K$^+$ buffer at pH 8, and 100 $\mu$M chlorhexidine and 5 nM valinomycin were added at 2 min. (b) (Proteo-)liposomes were diluted into Na$^+$ buffer at pH 8, and 100 $\mu$M chlorhexidine was added at 2 min. (c) (Proteo-)liposomes were diluted into K$^+$ buffer at pH 7, and 100 $\mu$M chlorhexidine and 5 nM valinomycin were added at 2 min. Liposome cartoons that illustrate experimental conditions are shown to the right. The fluorescence intensity measured at time zero is normalized to 1. Every experiment was done at least 3 times, and the error bars indicate standard deviations. wt, wild type.

induced a rapid efflux of protons, as indicated by the sharp increase in pyranine fluorescence. In comparison, significantly lower levels of alkalization were observed when adding 100 $\mu$M chlorhexidine to empty liposomes or proteoliposomes containing TqsA. The slow increase in the fluorescence signal of empty liposomes and the negative control might be attributed to a slight leakage of protons down their electrochemical gradient out of liposomes and the effect of chlorhexidine on naked liposomes (Fig. 2a, Fig. S8). These results indicate that PA2880 mediates chlorhexidine efflux by employing proton motive force.

Meanwhile, the transport activities of PA2880 variants were also examined by adding the same amount of chlorhexidine to the respective proteoliposomes. Compared to empty liposomes, only a slightly higher fluorescence change was observed in proteoliposomes containing E38Q only, and the proteoliposomes containing E106A behaved similarly to empty liposomes. In addition, no obvious fluorescence changes were observed in proteoliposomes containing D132A. For the other two variants, D83A and H101A, moderate fluorescence changes were observed in the lumen of proteoliposomes (Fig. 2a). These observations are consistent with the above-described *in vivo* drug susceptibility data.

The transport activities of PA2880 and its variants were then determined in the presence of $\Delta$pH (proton gradient) only. After generation of the proton gradient (acidic inside), the efflux of protons was initiated by adding 100 $\mu$M chlorhexidine to the system. pH changes in the lumen of proteoliposomes containing wild-type PA2880 and variants D83A and H101A were observed; although the changes in D83A and H101A proteoliposomes were slightly lower than those in wild-type PA2880 proteoliposomes, all of them were significantly above the background levels seen in empty liposomes or in proteoliposomes containing E38Q, E106A, or D132A (Fig. 2b). However, the fluorescence change was clearly lower than that observed when chlorhexidine was added to proteoliposomes after the generation of the electrochemical gradient.

Transport activity was also investigated in the presence of $\Delta\psi$ (membrane potential) only. The membrane potential was generated by diluting proteoliposomes, which were prepared in Na$^+$ buffer at pH 7.0, into K$^+$ buffer at pH 7.0 with valinomycin. The results indicate that the membrane potential affected PA2880 transport activity similarly to the proton gradient (Fig. 2c). Although not substantial, it can be seen that the electrical potential (membrane potential) did promote a more rapid PA2880-mediated chlorhexidine transport than the proton gradient.

**Chlorhexidine/H$^+$ antiport is an electrogenic process.** As chlorhexidine is cationic at physiological pH, we sought to determine whether the chlorhexidine/H$^+$ antiport process is electrogenic or electroneutral. To this end, we loaded PA2880 proteoliposomes with pyranine and examined the antiport process in the presence of symmetrical concentrations of chlorhexidine and protons. The membrane potential was set using the potassium ionophore valinomycin in the presence of a K$^+$ gradient, which could effectively "clamp" the voltage across the membrane (25, 26). No substrate movement occurs when the K$^+$ concentration is equal on both sides of the membrane (Fig. 3, black line). When an inwardly directed potassium gradient was applied (positive voltage inside), the observation of an increase in fluorescence indicated a net proton export (Fig. 3, red line). When the K$^+$ gradient was reversed, so that the liposomes were negatively charged inside, acidification of the liposome interior and net proton import were observed (Fig. 3, blue line). These experiments show that the membrane potential alone can drive chlorhexidine/H$^+$ antiport and determine its direction, supporting the notion that the transport activity of PA2880 is electrogenic (27–29).

On the other hand, more than one negatively charged residue (E38 [TM1], E106 [TM3], and D132 [TM4]) is essential for the PA2880 transport activity, as indicated by antimicrobial susceptibility testing (Table 1). The transport-negative PA2880 variants (E38Q, E106A, and D132A) bound chlorhexidine to considerable levels compared to the level of binding by the wild-type PA2880 (Table S3), suggesting that the transport defects observed in these variants are irrelevant for chlorhexidine binding. Altogether, these residues are not responsible for substrate binding. Considering the structural integrity of the variants (Fig. S6), we speculate that they might be located close together to bind protons.

**The oligomeric state of PA2880 in solution.** In this work, native mass spectrometry in the form of laser-induced liquid bead ion desorption mass spectrometry (LILBID-MS) (30, 31) was used to determine the oligomeric state of PA2880. Our results show that although signals from both monomeric and dimeric PA2880 can be observed, PA2880 mainly exists as a dimer ($\sim$70%) in solution (Fig. 4). However, as the proteins used in this study are eGFP fusion proteins, we needed to ensure that there was no effect (e.g., by steric hindrance) of the eGFP on the oligomeric state in solution. A comparison between wild-type PA2880 with and without a fused eGFP clearly showed very similar behaviors (Fig. 4b), demonstrating that the eGFP has no influence on the oligomeric state of PA2880.

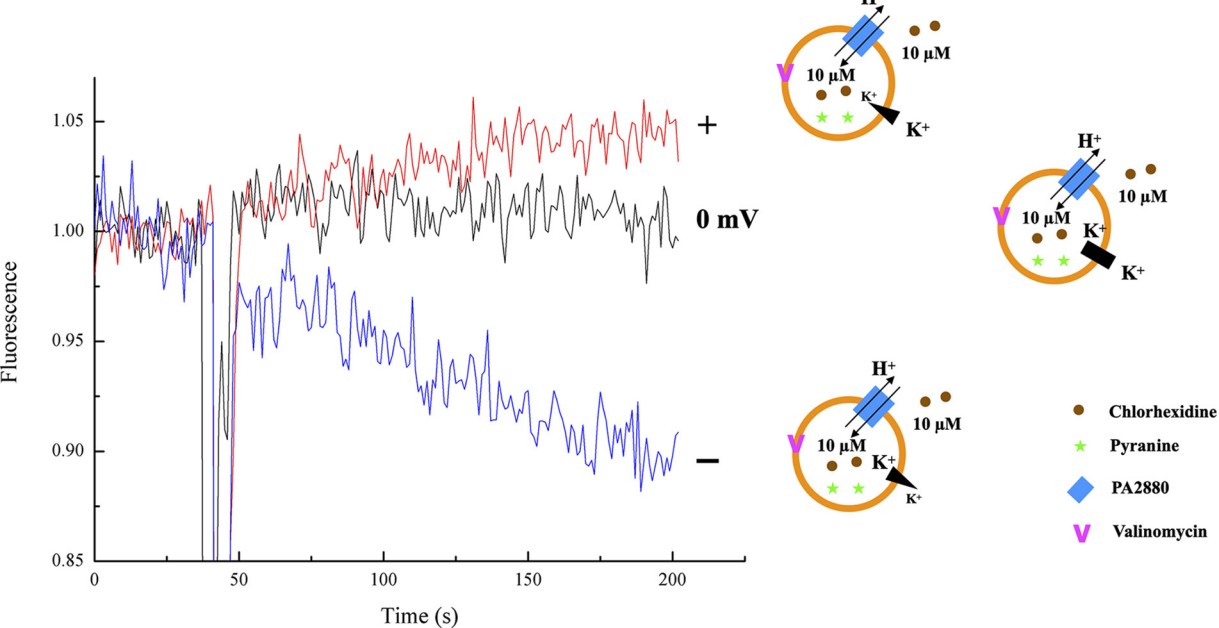

**FIG 3** Electrogenicity of chlorhexidine/H$^+$ antiport. Proton movement across PA2880 proteoliposomes was monitored by pyranine fluorescence. Proteoliposomes were prepared with equal internal and external pHs and chlorhexidine concentrations. Membrane potentials (approximately +36 mV, 0 mV, and −100 mV from top to bottom) were imposed by the addition of valinomycin to liposomes reconstituted with asymmetrical K$^+$ concentrations, indicated by the liposome cartoon immediately to the right of each trace. The detailed buffer conditions are listed in Table S4. Every experiment was repeated 3 times, and the results of one representative experiment are shown here.

LILBID-MS was also used to investigate the effect of pH on the protein's oligomeric state. The results showed that PA2880 with and without eGFP behaved very similarly and mainly existed as a dimer (~70%) over the pH range from 5 to 9, implying that the PA2880 dimer is stable in the whole pH range. There is a possibility that the ratio of dimers is even higher in solution, as dissociation of the dimer due to the laser power used in LILBID is possible. Meanwhile, we also carried out analogous experiments with the E38Q variant and found that the E38Q variant is also mainly present as a dimer, suggesting a negligible effect of E38 on protein dimer formation.

In addition, we performed the measurement in the presence of the substrate chlorhexidine. The LILIBID-MS results showed that chlorhexidine was bound to PA2880, and a significant increase in the dimer population was observed when chlorhexidine was added to the protein. To explore this observation further, we also performed the measurements with an increasing substrate/protein molar ratio up to 8. As shown by the results in Fig. 4, the addition of chlorhexidine increased the dimer formation. Taken together, the results showed that PA2880 should function as a dimer and that the assembly of the dimer is supported by the binding of chlorhexidine.

In conclusion, the study presented here demonstrates that PA2880, the PACE transporter from *P. aeruginosa*, could confer resistance to chlorhexidine. Our data suggest that PA2880 is a proton-dependent transporter and that protons might be bound by several highly conserved residues (E38, E106, and D132). In addition, PA2880 should function as a dimer, and its dimerization is stable over a wide range of pHs from 5 to 9. The phylogenetic analysis clearly shows a difference between PA2880 and AceI, although both of them could mediate the efflux of chlorhexidine. Future structural and functional analyses are expected to shed light on the protein's assembly properties and its transport mechanism. It will be also worth investigating the proteins that are unresponsive to chlorhexidine to advance our understanding of the PACE family transporters.

## MATERIALS AND METHODS

**Genome data.** Genomic data used in this study were obtained from the National Center for Biotechnology Information reference sequence database (version 205, accessed 5 March 2021). The database was filtered for

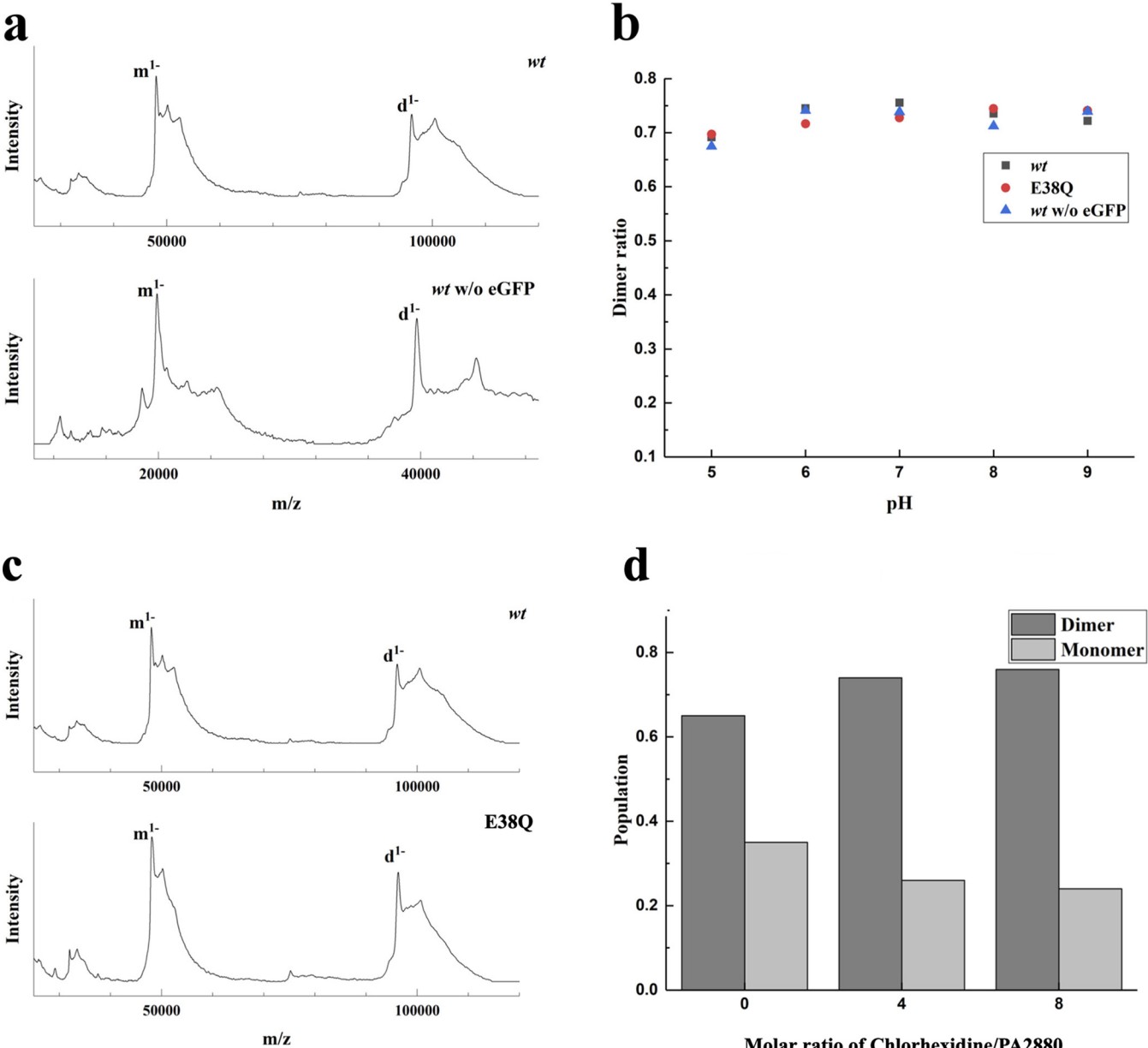

**FIG 4** Native MS investigation of the dimerization of PA2880. (a) The LILBID spectra of the wild-type (wt) PA2880 with/without fusion of eGFP. All measurements were performed at pH 8. (b) The pH dependence of the dimerization of all aforementioned PA2880 variants. No strong effects of pH on protein dimerization were detected. The dimer ratio was calculated from deconvoluted zero-charge spectra, taking the overlap of different charge states of monomer and dimer into account. (c) The spectra of the PA2880 wild type and the E38Q variant. All measurements were performed at pH 8. (d) PA2880 dimer population in the presence of chlorhexidine, indicating that chlorhexidine strengthens the dimer formation.

genome assemblies taxonomically assigned to the phylum *Proteobacteria*. For the final set of 1,363 genomes, only assemblies with a complete or nearly complete ("chromosome") assembly status that, in addition, were categorized by the NCBI as a representative or reference genome of their species were selected. In the case of multiple reference/representative genomes per species identified by identical NCBI taxonomy identifiers, we selected one at random. Finally, as a reference genome for subsequent studies, the current type strain of *Acinetobacter baumannii* (American Type Culture Collection strain ATCC 19606; NCBI assembly accession number GCF_000737145.1) was added to the set.

**Phylogenetic profiling.** We identified orthologs of the two *A. baumannii* PACE family transporters (AceI [WP_002010078.1] and AS_1503 in *A. baumannii* ATCC 17979 [WP_001161759.1]) with fdog (version 0.0.36, https://github.com/BIONF/fDOG) (32) using *A. baumannii* ATCC 19606 (GCF_000737145.1) as the reference taxon. The pHMM for the final ortholog search was trained with sequences from the genus *Acinetobacter* only (option maxdist=genus). We used the same approach to determine the phylogenetic profiles of the LysR family transcriptional regulators (accession numbers WP_001102208.1 and WP_001161759.1) that flank AceI and the PACE efflux transporter (WP_001161759) in *A. baumannii* ATCC

19606, respectively. The conservation of the gene pairing across the proteobacterial diversity was assessed with an in-house script.

**Phylogeny reconstruction.** For phylogenetic-tree reconstruction, we aligned the sequences of the individual orthologous groups with Muscle (33), using the default options. Alignment columns with more than 50% gaps were subsequently removed using a custom Perl script. The post-processed alignment served as the input for a maximum-likelihood tree reconstruction with IQ-Tree version 2.0.3 (34) using the LG+G+I+F model of sequence evolution. Statistical branch support was assessed with 1,000 bootstrap replicates using the ultrafast bootstrapping algorithm (35) implemented in IQ-Tree. The trees were visualized in ITol (36).

**Construction of the expression vector.** The PA2880 gene (NCBI Gene ID 882666) was amplified from genomic DNA of *Pseudomonas aeruginosa* strain PAO1 by PCR. The enhanced green fluorescence protein (eGFP) fragment (A206K variant, in order to minimize its dimerization tendency [37]) was also amplified by PCR and cloned into the vector pL2020 (38) using the InFusion ligation-independent cloning method, and then the PA2880 gene was cloned into the above-described ligated vector, resulting in the expression construct pL2020-PA2880-eGFP, which encodes PA2880 with a tobacco etch virus (TEV) protease site between PA2880 and eGFP and a Strep-tag fused to the C terminus.

**Electroporation of *Pseudomonas stutzeri*.** Electrocompetent cells of *P. stutzeri* were prepared according to the procedure of Choi et al. (39), with slight modifications. Briefly, 1 mL of cells in the early stationary phase (optical density at 600 nm [$OD_{600}$] of 1.5 to 2.0) from cultures grown in lysogeny broth (LB) medium were harvested by centrifugation at $16,000 \times g$ and washed with 1 mL sterilized 300 mM sucrose three times. Cells resuspended in 100 $\mu$L 300 mM sucrose were mixed with 100 ng plasmid DNA in a 1-mm electroporation cuvette. High-voltage electroporation was performed using a Bio-Rad Gene Pulser at 25 $\mu$F, 200 $\Omega$, and 2.5 kV. After applying the pulse, 1 mL of SOC (super optimal broth with catabolite repression) medium was added immediately and the cells were transferred to a culture tube and incubated at 37°C for 1 h. Cells were plated on LB agar plates with chloramphenicol and incubated at 32°C for 48 to 72 h.

**Protein expression and purification.** Initially, the overexpression of PA2880 was performed in *E. coli* BL21 cells transformed with the pTTQ-PA2880-eGFP plasmid. However, the yield was not satisfying. According to the research in our group (38), *Pseudomonas stutzeri* is a better host for membrane proteins than *E. coli*. Therefore, for heterologous overexpression, *P. stutzeri* cells transformed with pL2020-PA2880-eGFP plasmid were grown at 32°C in LB medium supplemented with 34 $\mu$g/mL chloramphenicol to an $OD_{600}$ of 0.5. Production of PA2880 was induced by the addition of 0.02% (wt/vol) L-arabinose, and incubation was continued at 32°C for 3 h. Cells were harvested by centrifugation at $10,000 \times g$ for 15 min, frozen in liquid nitrogen, and stored at −80°C until use.

All steps of membrane preparation and protein purification were performed at 4°C. Cells were resuspended in resuspension buffer (50 mM Tris-HCl [pH 8], 300 mM NaCl, 2 mM phenylmethanesulfonylfluoride [PMSF], 20 $\mu$g/mL DNase I) in a ratio of 7 mL buffer per 1 g cells and were disrupted three times by passing through a precooled microfluidizer (Microfluidics, USA) at 8,000 lb/in². After centrifugation at $27,000 \times g$ for 60 min to remove unbroken cells and cell debris, membranes were collected by centrifugation at $144,000 \times g$ for 2.5 h. Crude membranes were resuspended in solubilization buffer (50 mM Tris-HCl [pH 8], 300 mM NaCl) and membrane proteins were solubilized by moderate stirring with 1% (wt/vol) *n*-dodecyl $\beta$-D-maltoside (DDM) for 1.5 h. The insoluble membrane fraction was removed by centrifugation at $222,000 \times g$ for 1 h, and the supernatant containing solubilized membrane proteins was loaded onto a Strep XT high-capacity column (IBA, Germany) equilibrated with 50 mM Tris-HCl (pH 8), 300 mM NaCl, 0.05% (wt/vol) DDM. After extensive washing with equilibration buffer, the bound proteins were eluted using the equilibration buffer supplemented with 50 mM biotin. The eluted fractions were pooled and concentrated using Amicon Ultra-15 concentrators (50,000 molecular weight cutoff [MWCO]; Merck Millipore, Germany). The concentrated protein was purified further by size exclusion chromatography using a Superose 6 increase 10/300 column (GE Healthcare, USA) equilibrated in 50 mM Tris-HCl (pH 8), 300 mM NaCl, 0.05% (wt/vol) DDM. Peak fractions containing PA2880-eGFP proteins were collected, concentrated, and stored at −80°C. Protein concentration was determined using the bicinchoninic acid (BCA) assay (Thermo Scientific Pierce, USA).

**Antimicrobial susceptibility test.** MIC determinations were conducted in Mueller-Hinton (MH) medium by the broth dilution method as described previously (40). MICs of *E. coli* strain BL21 carrying pTTQ18-based plasmids were determined in the presence of 0.05 mM isopropyl $\beta$-D-1-thiogalactopyranoside (IPTG). Cell growth was evaluated by visual inspection after 20 h of incubation at 37°C.

**Reconstitution of protein PA2880 and its variants.** Proteoliposomes were prepared as described by Hassan et al. (14), with some modifications. Liposomes were prepared using the *E. coli* polar lipid extract (Avanti Polar Lipids). Amounts of 25 mg of lipids were dried under nitrogen gas from the chloroform solution and then suspended in 2.5 mL of reconstitution buffer (25 mM HEPES-NaOH [pH 7], 200 mM NaCl), and the lipid suspension was extruded 21 times through a 400-nm filter to form liposomes. Five-hundred-microliter samples containing 300 $\mu$L of preformed liposomes, 1 mM pyranine, 1.1% *n*-octylglucoside (OG) to destabilize the liposomes, and 0.3 mg of purified PA2880 or its variants were prepared. For control liposomes, the same volume of purification buffer or protein TqsA was added. The samples were incubated at room temperature for 15 min and then diluted 20 times with cold reconstitution buffer supplemented with 1 mM pyranine to dilute the OG to a concentration below its critical micelle concentration and to keep the pyranine concentration constant inside the proteoliposomes. The diluted sample was ultracentrifuged at $222,000 \times g$ for 1 h to collect the proteoliposomes. The proteoliposomes were resuspended in 500 $\mu$L of reconstitution buffer and then subjected to PD Minitrap G-25 desalting columns (Thermo Fisher Scientific, USA) preequilibrated with reconstitution buffer to remove external pyranine. The proteoliposomes were collected by ultracentrifugation

at 222,000 × $g$ for 1 h and resuspended with 100 $\mu$L reconstitution buffer prior to conducting the transport experiments.

**Chlorhexidine/H$^+$ antiport assays using proteoliposomes containing pyranine.** The proteoliposomes were diluted 100-fold into different buffers to generate proton and/or voltage gradients across the membrane prior to the addition of chlorhexidine diacetate (referred to as chlorhexidine herein) and valinomycin. To generate a proton electrochemical potential (acidic and positive inside), the proteoliposomes were diluted into 25 mM HEPES-NaOH (pH 8.0), 200 mM KCl, with the addition of 5 nM valinomycin and substrate when measuring the transport activity. To generate a chemical proton gradient ($\Delta$pH, acidic inside), the proteoliposomes were diluted into 25 mM HEPES-NaOH (pH 8.0), 200 mM KCl, and no valinomycin was added. To generate an electrical gradient only ($\Delta\psi$, positive inside), the proteoliposomes were diluted into 25 mM HEPES-NaOH (pH 7.0), 200 mM KCl, with the addition of 5 nM valinomycin and substrate when measuring the transport activity. Chlorhexidine was added to a concentration of 100 $\mu$M. The excitation maximum of pyranine shifted from 400 nm to 450 nm with increasing pH, while the emission maximum was stable at $\sim$515 nm. Therefore, a fluorescence proportion of 450 nm/515 nm was used to monitor the pH change in the lumen of proteoliposomes, and an increase of this ratio was indicative of an increase of the internal pH.

**Electrogenicity experiments of PA2880.** The electrogenicity experiments were performed according to the method described by Fitzgerald et al. (29), with slight modifications. After the affinity chromatography, the protein was purified further by size exclusion chromatography using a Superose 6 increase 10/300 column equilibrated in 25 mM HEPES-KOH, pH 7, 50 mM KCl, 0.05% (wt/vol) DDM. Then, protein PA2880 was reconstituted into liposomes as described above, with slight modifications. The proteoliposomes were prepared with 25 mM HEPES-KOH, pH 7, 50 mM KCl, 1 $\mu$M pyranine, and 10 $\mu$M chlorhexidine. Choline chloride was added to maintain osmotic pressure. For electrogenicity experiments, the proteoliposomes were diluted 100-fold into external buffer, which contained 25 mM HEPES-KOH, pH 7, 1 to 200 mM KCl to establish a potassium gradient (Table S4), choline chloride to maintain osmotic pressure, 10 $\mu$M chlorhexidine, and 5 nM valinomycin to set the membrane potential. Membrane potential was calculated from the K$^+$ gradient according to the Nernst potential. Pyranine fluorescence was monitored as an indication of proton transport as before.

**Native MS/LILBID-MS.** For native mass spectrometry, the samples were analyzed by laser-induced liquid bead ion desorption mass spectrometry (LILBID-MS). A more detailed description of LILBID is given in references 31 and 41. Briefly, a piezo-driven droplet generator (MD-K-130; Microdrop Technologies GmbH, Norderstedt, Germany) was used to produce droplets of 30 $\mu$m in diameter with a frequency of 10 Hz at a pressure of 100 millipascals. Before loading the samples, proteins were buffer exchanged into 200 mM ammonium acetate at desired pHs and chlorhexidine concentrations, with 0.02% (wt/vol) DDM. Then, samples were loaded directly into the droplet generator and the generated droplets were subsequently transferred to high vacuum and irradiated by a mid-infrared (IR) laser directly in the ion source. The laser employed was a neodymium-doped yttrium aluminum garnet (Nd:YAG) laser operating at 10 Hz, the wavelength being tuned by a LiNbO$_3$ optical parametric oscillator to 2.94 $\mu$m $\pm$ 5 nm, the absorbing wavelength of the symmetric and asymmetric O-H stretching vibration of water. The pulse length was 6 ns with a maximum pulse energy of 12 mJ. The laser power was measured by an optical power meter (PM100D; Thorlabs, Munich, Germany).

Droplet irradiation leads to an explosive expansion of the droplet containing the sample, and solvated ions were released and analyzed in an in-house-built time-of-flight setup operating at 10$^{-6}$ millipascals. The ion source is based on a Wiley-McLaren-type design. The ions are accelerated into the grounded flight tube and guided toward the detector via a reflectron. The detector setup is based on a Daly-type detector optimized for the detection of high $m/z$ ions. The voltage of the first (repeller) and second plate was set to $-4$ kV in the ion source. The third plate was grounded. The repeller was pulsed to $-6.6$ kV for 370 $\mu$s after droplet irradiation. The pulse was applied between 2 and 20 $\mu$s after the droplet irradiation (delayed extraction time). The einzel lenses were set to $-3.0$ kV. The reflectron was set to $-7.2$ kV. Postacceleration was set to $+17$ kV at the microchannel plate (MCP) impact surface. Spectrum processing was done by using the software *Mass*ign (42) based on LabVIEW. To account for the overlap of different charge states in the spectra, the spectrum analysis software UniDec (43) was used to deconvolute the spectra to a zero-charge spectrum (41), from which the monomer/dimer ratio of protein PA2880 in solution was calculated.

**Polyacrylamide gel electrophoresis and Western blotting.** Sodium dodecyl sulfate-polyacrylamide gel electrophoresis (SDS-PAGE) was performed using 4 to 12% Bis-Tris NuPAGE gels (Invitrogen, USA) followed by Coomassie blue staining. The Strep-tagged PA2880 was immunodetected using a monoclonal anti-Strep alkaline phosphatase-conjugated antibody (Sigma, USA), following the manufacturer's instructions. The NBT-BCIP (nitro-blue tetrazolium–5-bromo-4-chloro-3-indolylphosphate) system was used to detect the alkaline phosphatase activity.

## SUPPLEMENTAL MATERIAL

Supplemental material is available online only.
**SUPPLEMENTAL FILE 1**, PDF file, 6.9 MB.

## ACKNOWLEDGMENTS

H.M. and J.W. supervised the research. H.M. and J.Z. designed the research. J.Z. prepared samples and performed research. N.H. and N.M. did the native mass spectrometry analysis. J.Z.

and N.H. analyzed the data. R.K. performed additional transport assays in the reconstituted system. B.D. and I.E. performed the phylogenetic analyses. J.Z., N.H., and I.E. wrote the manuscript.

The BW25113 (ΔacrB ΔemrE ΔmdfA) strain was kindly provided by Klaas M. Pos from the Institute of Biochemistry, Goethe University Frankfurt.

This work was supported by the Max Planck Society, the Center of Excellence (Macromolecular Complexes) Frankfurt, the international academic exchange fund of the Graduate School of Tianjin University, and a grant from the German Research Foundation (DFG) in the scope of the Research Group FOR2251 Adaptation and Persistence of *A. baumannii* (grant number EB-285-2/2).

We declare no competing interests.

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
