## [Reviewer comments · Microbiology Spectrum]

Microbiology Spectrum

Assembly and functional role of the PACE transporter PA2880 from *Pseudomonas aeruginosa*

Jiangfeng Zhao, Nils Hellwig, Bardya Djahanschiri, Radhika Khera, Nina Morgner, Ingo Ebersberger, Jingkang Wang, and Hartmut Michel

Corresponding Author(s): Hartmut Michel, Max Planck Institute of Biophysics

Review Timeline:

Submission Date:	September 2, 2021
Editorial Decision:	October 18, 2021
Revision Received:	March 10, 2022
Accepted:	March 12, 2022

Editor: Ayush Kumar

Reviewer(s): The reviewers have opted to remain anonymous.

Transaction Report:

DOI: <https://doi.org/10.1128/spectrum.01453-21>

October 18, 2021

Prof. Hartmut Michel
Max Planck Institute of Biophysics
Molecular Membrane Biology
Max-von-Laue-Str. 3
Frankfurt am Main D-60438
Germany

Re: Spectrum01453-21 (Assembly and functional role of the PACE transporter PA2880 from *Pseudomonas aeruginosa*)

Dear Prof. Hartmut Michel:

Thank you for submitting your manuscript to Microbiology Spectrum. It has now been reviewed by two experts. They both express enthusiasm about your work however are suggesting a number of modifications, including some additional experiments. I believe the modifications suggested by reviewers are important to address before I can consider your manuscript for publication in Spectrum. When submitting the revised version of your paper, please provide (1) point-by-point responses to the issues raised by the reviewers as file type "Response to Reviewers," not in your cover letter, and (2) a PDF file that indicates the changes from the original submission (by highlighting or underlining the changes) as file type "Marked Up Manuscript - For Review Only". Please use this link to submit your revised manuscript - we strongly recommend that you submit your paper within the next 60 days or reach out to me. Detailed information on submitting your revised paper are below.

Link Not Available

Sincerely,

Ayush Kumar

Journals Department
Reviewer comments:

Reviewer #1 (Comments for the Author):

To date few studies have been performed into the function of PACE family transport proteins, and the majority of these have focussed on only a few members. Therefore, this work is very welcome and important. I have a number of comments and suggestions. Some of these are simple typographical suggestions, whereas others are more important.

Line 26. Most recently of what? Reword to "The recently identified ..."?

Lines 53-54. The references here don't fit with the idea of "Accumulating evidence", since both are from 2004.

Lines 56-57. They may be the most recently identified bacterial drug efflux transporters, but probably not "the latest identified secondary active transporters".

Line 70. It was suggested, not reported.

Line 99. "we determined the phylogenetic profile of Acel..." should be something like "we identified the predicted orthologs of Acel..."

Line 110. What is meant by "... a group of highly derived members of this subfamily"?

Line 112. With 2.49 substitutions per site there would be no phylogenetic information remaining, in which case the bootstrap support would be expected to be low. Is it per site or per x nt?

Lines 94-141. This is a very long paragraph and lacks clarity. The important point is in the last sentence Lines 138-141. I suggest condensing this considerably.

Line 161. "Altogether, PA2880 mediates the efflux of chlorhexidine." At this point of the manuscript, it can only be said that PA2880 mediates resistance to chlorhexidine. The transport experiments are presented later.

Lines 166-174. Amino acid sequence motifs have been identified in PACE pumps (<https://doi.org/10.1016/j.resmic.2018.01.001>). Could the amino acid residues referenced in this section be put into the context of these motifs?

Line 201. The fluorescence changes associated with the E38A mutant appear slightly above background in empty liposomes.

Line 224-249 and Fig 3. The fluorescence changes are very low in these experiments. Were empty liposomes or proteoliposomes carrying an inactive mutant also used as a control to show that the changes are not due only to the leakage of protons down the electrical gradient?

Line 225. From my understanding, chlorhexidine is divalent. The charged groups are also physically separated on opposite sides of the molecule and it is possible that two translocation cycles may occur during efflux via some pumps, eg.

<https://doi.org/10.1038/ncomms5615>. Some discussion of these points could be included in the manuscript. This is very important as it also impacts the conclusion that more than four H⁺ are exchanged for each substrate (lines 33, 88, 240).

Line 307. Rather than "The phylogenetic profiles of the two ...", I think the authors mean "The predicted orthologs of the two ... "

Lines 413 and 417. The schematic in Figure 4 suggests that 10 μ M chlorhexidine was used in these experiments, not 1 μ M.

Supplement Table S2 "BW25113 (Δ acrB, Δ emfA, Δ emrE)" should be BW25113 (Δ acrB, Δ mdfA, Δ emrE)

Supplement Figure S4. What is the molecular weight of PA2880? Are the authors confident of the monomer/dimer designations?

Supplement Figure S5. The purified PA2880 migrates at 150+ kDa relative to the markers in BN-PAGE. How do the authors interpret this with respect to the oligomeric state of the protein?

Reviewer #2 (Comments for the Author):

Zhao and coworkers present a study on the *Pseudomonas aeruginosa* protein PA2880, which belongs to the PACE family of transporters. Firstly, they present a phylogenetic analysis of the protein in order to determine its evolutionary relationship to other members of the family. The roles of conserved residues in the function of the protein are investigated by introducing mutations and comparing the function of the mutated proteins to the wildtype in minimum inhibitory concentration assays, transport experiments with protein reconstituted into liposomes and native mass spectrometry experiments. Understanding the function of PA2880 is a useful contribution to the field as little is known about the function of any proteins of the PACE family beyond the representative protein Acel. The protein of interest is also found in a pathogen so unravelling its mechanism would be relevant for understanding and combating antimicrobial resistance. However, the data in this study is not strong enough to support the conclusions made and further investigations should be carried out to complete this work.

General remarks:

1. The authors conclude that PA2880 cannot transport cadaverine based on the MIC assays. However, in previously reported transport assays (in reference 14 of the manuscript) with reconstituted Acel, the cadaverine concentration used at pH 7.0 was 100 mM, which is far above the concentrations used in the MIC assays, so the observed effect could simply be a case of the cadaverine concentration used being too low rather than it not being a substrate.
2. The data clearly shows that the W74A mutation perturbs the SDS-PAGE and SEC profiles of the protein as compared to the wild-type and other mutants. This raises concerns about the integrity of the W74A mutant protein, so this mutant should not be discussed further and neither should conclusions be made about this amino acid substitution from the experimental data.
3. Supplemental data 1 and 2 should contain the sequence alignments but they do not display as intended - this needs to be addressed.
4. In reference 14, which is the paper the current liposome transport assays are based on, it is noted that chlorhexidine may damage naked proteoliposomes. This puts the findings from the liposome assays under question. These conclusions need to be justified by performing controls (e.g. reconstituting a protein which does not transport chlorhexidine), statistics on the data to determine the significance of the difference between wild-type and mutant proteins or performing an independent experiment to corroborate the findings (e.g. with radiolabelled chlorhexidine)
5. I would like to see the authors' working in more detail in relation to the calculation of more than 4 protons being exchanged for every molecule of chlorhexidine.

Following from the above point, I am unsure that quantitative observations could be made on the basis of the data from the liposome assays. Furthermore, chlorhexidine may be getting (de)protonated as it moves between environments at different pH, making a calculation which does not take this into account unreliable. The idea that four protons are transported, but nonetheless transport is primarily dependent on charge, not pH, seems surprising and should be supported by reference to other, similar, examples.

6. The authors state that LILBID-MS, which they used, is comparable to nESI-MS, which is what Bolla et al (2020; cited as 'Robinson et al' in the manuscript) used to study Acel. However, there is no direct comparison made between the two studies. The authors should run a LILBID-MS experiment with Acel as Bolla and colleagues did to directly compare their findings. This would alleviate doubts about the effects observed being due to a discrepancy between the two methods, especially as the oligomeric states that Zhao and colleagues report are very different from those published about Acel - and yet the proteins are nonetheless clearly part of the same

structural family. An effort to reconcile the Bolla et al. observation that the PACE proteins dimerise as a function of pH with the obligate dimer found here would be important.

7. On lines 32-33 in the Abstract the authors state that the dimerization of PA2880 is essential for its function (based on the mass spectrometry experiments). However, these experiments were only performed on the protein on its own. An experiment with the substrate chlorhexidine would shed more light on what the relationship between the substrate and the oligomerisation state is. No direct comparison can be drawn between the mass spectrometry data and the liposome assays since the protein is solubilised in DDM for LILBID-MS.

8. The English language in the manuscript is poor throughout (e.g. in the 'Importance' statement the first sentence on lines 39-40 refers to a pathogen being related to infections, which is the wrong word choice).

Staff Comments:

Preparing Revision Guidelines

Please return the manuscript within 60 days; if you cannot complete the modification within this time period, please contact me. If you do not wish to modify the manuscript and prefer to submit it to another journal, please notify me of your decision immediately so that the manuscript may be formally withdrawn from consideration by Microbiology Spectrum.

Zhao and coworkers present a study on the *Pseudomonas aeruginosa* protein PA2880, which belongs to the PACE family of transporters. Firstly, they present a phylogenetic analysis of the protein in order to determine its evolutionary relationship to other members of the family. The roles of conserved residues in the function of the protein are investigated by introducing mutations and comparing the function of the mutated proteins to the wild-type in minimum inhibitory concentration assays, transport experiments with protein reconstituted into liposomes and native mass spectrometry experiments.

Understanding the function of PA2880 is a useful contribution to the field as little is known about the function of any proteins of the PACE family beyond the representative protein Acel. The protein of interest is also found in a pathogen so unravelling its mechanism would be relevant for understanding and combating antimicrobial resistance. However, the data in this study is not strong enough to support the conclusions made and further investigations should be carried out to complete this work.

General remarks:

- The authors conclude that PA2880 cannot transport cadaverine based on the MIC assays. However, in previously reported transport assays (in reference 14 of the manuscript) with reconstituted Acel, the cadaverine concentration used at pH 7.0 was 100 mM, which is far above the concentrations used in the MIC assays, so the observed effect could simply be a case of the cadaverine concentration used being too low rather than it not being a substrate.
- The data clearly shows that the W74A mutation perturbs the SDS-PAGE and SEC profiles of the protein as compared to the wild-type and other mutants. This raises concerns about the integrity of the W74A mutant protein, so this mutant should not be discussed further and neither should conclusions be made about this amino acid substitution from the experimental data.
- Supplemental data 1 and 2 should contain the sequence alignments but they do not display as intended - this needs to be addressed.
- In reference 14, which is the paper the current liposome transport assays are based on, it is noted that chlorhexidine may damage naked proteoliposomes. This puts the findings from the liposome assays under question. These conclusions need to be justified by performing controls (e.g. reconstituting a protein which does not transport chlorhexidine), statistics on the data to determine the significance of the difference between wild-type and mutant proteins or performing an independent experiment to corroborate the findings (e.g. with radiolabelled chlorhexidine)
- I would like to see the authors' working in more detail in relation to the calculation of more than 4 protons being exchanged for every molecule of chlorhexidine. Following from the above point, I am unsure that quantitative observations could be made on the basis of the data from the liposome assays. Furthermore, chlorhexidine may be getting (de)protonated as it moves between environments at different pH, making a calculation which does not take this into account unreliable. The idea that four protons are transported, but nonetheless transport is primarily dependent on charge, not pH, seems surprising and should be supported by reference to other, similar, examples.

- The authors state that LILBID-MS, which they used, is comparable to nESI-MS, which is what Bolla *et al* (2020; cited as 'Robinson *et al*' in the manuscript) used to study Acel. However, there is no direct comparison made between the two studies. The authors should run a LILBID-MS experiment with Acel as Bolla and colleagues did to directly compare their findings. This would alleviate doubts about the effects observed being due to a discrepancy between the two methods, especially as the oligomeric states that Zhao and colleagues report are very different from those published about Acel – and yet the proteins are nonetheless clearly part of the same structural family. An effort to reconcile the Bolla *et al.* observation that the PACE proteins dimerise as a function of pH with the obligate dimer found here would be important.
- On lines 32-33 in the Abstract the authors state that the dimerization of PA2880 is essential for its function (based on the mass spectrometry experiments). However, these experiments were only performed on the protein on its own. An experiment with the substrate chlorhexidine would shed more light on what the relationship between the substrate and the oligomerisation state is. No direct comparison can be drawn between the mass spectrometry data and the liposome assays since the protein is solubilised in DDM for LILBID-MS.
- The English language in the manuscript is poor throughout (e.g. in the 'Importance' statement the first sentence on lines 39-40 refers to a pathogen being related to infections, which is the wrong word choice).

Comments to editor:

We would recommend that the manuscript be rejected if the English language is still unacceptable upon revision.

Reviewed by Maria Nikolova (graduate student) and Professor Adrian Goldman (PhD advisor)

To the editors of the journal Spectrum

Prof. Dr. Dr. Hartmut Michel
Director

Tel.: 0049 - (0)69 - 6303 - 1001

Fax: 0049 - (0)69 - 6303 - 1002

hartmut.michel@biophys.mpg.de

March 4th, 2022

Dear Dr. Kumar,

We would like to thank all reviewers for their insightful comments and valuable suggestions, which helped us to improve the quality of our manuscript (Spectrum01453-21: Assembly and functional role of the PACE transporter PA2880 from *Pseudomonas aeruginosa*). We would like to thank you for giving us the opportunity to resubmit our revised manuscript. In this revised manuscript, we have followed the reviewers' suggestions and a revision has been made accordingly. All changes are marked in red in the revised manuscript. Our point-to-point responses are given below.

Reviewer #1:

To date few studies have been performed into the function of PACE family transport proteins, and the majority of these have focussed on only a few members. Therefore, this work is very welcome and important. I have a number of comments and suggestions. Some of these are simple typographical suggestions, whereas others are more important.

We appreciate reviewer's supportive and constructive comments. Our responses to your concerns are given below.

Line 26. Most recently of what? Rerword to "The recently identified ..."?

Following the reviewer's suggestion, we rephrased it to "The recently identified...".

Lines 53-54. The references here don't fit with the idea of "Accumulating evidence", since both are from 2004.

We corrected our description in the revised manuscript to "Several studies..."

Lines 56-57. They may be the most recently identified bacterial drug efflux transporters, but probably not "the latest identified secondary active transporters".

To be more precise, we rephrased this sentence to "...are the most recently identified bacterial drug efflux transporters".

Line 70. It was suggested, not reported.

The word was corrected as reviewer suggested.

Line 99. "we determined the phylogenetic profile of AceI..." should be something like "we identified the predicted orthologs of AceI..."

Thank you very much for this suggestion. However, the term 'phylogenetic profile' is defined as follows: "The phylogenetic profile of a protein stores information about the presence and the absence of that protein (i.e. its orthologs) in a set of genomes" (https://www.ebi.ac.uk/ols/ontologies/mi/terms?obo_id=MI%3A0085; Pelgrini et al. PNAS 1999; 10.1073/pnas.96.8.4285). We therefore trust that the phrase here is correct as it stands, because we first specify that we analyse 1364 strains and find that many of these do not harbour the protein. From this profile, we then extract the 399 orthologs from 383 taxa.

Line 110. What is meant by "... a group of highly derived members of this subfamily"?

We apologise for the lab jargon... We meant to say that AceI sequences from Acinetobacter form a well separated clade that is connected via a long internal branch to the remainder of the tree. Using branch lengths as a measure of evolutionary time, which is measured here in substitutions per site, we show that the AceI clade is only very distantly related to any other sequence in our data set, and in particular to PA2880. We have re-written the paragraph to enhance clarity.

Line 112. With 2.49 substitutions per site there would be no phylogenetic information remaining, in which case the bootstrap support would be expected to be low. Is it per site or per x nt?

Branch lengths are indeed given in substitutions per site and upfront, the branch length is, if at all, only loosely correlated with the bootstrap support of an internal branch (split) in the tree. In essence, branch lengths are a poor proxy of sequence similarity. The alignment below shows that the two sequences, PA2880 and AceI, are similar enough to make their alignment straightforward. Moreover, the similarity extends over the entire sequence, apart from the terminal gap.

CLUSTAL 2.1 multiple sequence alignment

```
PSEAE208964@1|NP_251570.1|1      MTIQPDFNDEPGAFAMTHHTALDKTLKERIFHALAFEGGLAVLLTAPVLS
ACIBA8575584@1|WP_002010078.1|  MLIS-----KRLIHAIISYEGILLVIAIALS
* *                               *.*:**:**: ** : : * .**

PSEAE208964@1|NP_251570.1|1      LVMNKPLAHMGALTLMFSTVAMLWNMLFNSLFDRAQRMGFQRTLQVRVL
ACIBA8575584@1|WP_002010078.1|  FIFDMPMEVTGLGVFMAVSVFVWNI FNHYFEKVEHKFNWERTIPVRIL
: : : * : * : : : : * : : : : * : : : : * : : : : * : : : *

PSEAE208964@1|NP_251570.1|1      HAMLFELGLIVLVPLAANWLSIGLVEAFLDMLLFFLPYTMAFNWSY
ACIBA8575584@1|WP_002010078.1|  HAIGFEGGLLIATVPMIAYMMQMTVIDAFILDIGLTLCLLVYTFIFQWCY
```

```

** : ** *::: . ** : * : :. : :::**:* * : * ** : * : * . *
PSEAE@208964@1|NP_251570.1|1      DVLRLRVESRQAKAAGCDAG
ACIBA@575584@1|WP_002010078.1|    DHIEDKFFPN--AKAASLH--
* : . : : . . * * * . .

```

We want to give the reviewer a brief intuition of where the long branches come from. We model sequence evolution under a rate-heterogeneous model. In a nutshell, we assign each position in the alignment its own substitution rate differentiating between quickly and slowly evolving positions across an alignment. Sites that are highly conserved across the sequences in the alignment will get assigned a low substitution rate. If the *Acinetobacter* sequences carry a substitution at such a site, the model will have to assume that a considerable amount of time – measured in substitutions per site - has passed on the lineage leading to these sequences such that such a substitution becomes likely. The tree topology, however, is jointly determined by all positions in the alignment. During bootstrapping, only the tree topology, i.e. the splits in the tree, are evaluated, but not the branch length.

Eventually, why there are many splits in the tree only poorly supported by the bootstrap analysis? This is essentially due to the small alignment length that does not carry sufficient phylogenetic information to confidently resolve all splits. However, we would like to stress that the essential splits in the tree, i.e. those that group PA2880 with sequences from the beta-proteobacteria instead with the AceI clade achieve high bootstrap support.

Lines 94-141. This is a very long paragraph and lacks clarity. The important point is in the last sentence
Lines 138-141. I suggest condensing this considerably.

We agree with the reviewer that this paragraph was a bit over the top. We have condensed it to convey the main message to the reader. The remainder of the information was moved as Supplementary Text into the supplement.

Line 161. "Altogether, PA2880 mediates the efflux of chlorhexidine." At this point of the manuscript, it can only be said that PA2880 mediates resistance to chlorhexidine. The transport experiments are presented later.

We rephrased the sentence to “Altogether, PA2880 mediates resistance to chlorhexidine”.

Lines 166-174. Amino acid sequence motifs have been identified in PACE pumps (<https://doi.org/10.1016/j.resmic.2018.01.001>). Could the amino acid residues referenced in this section be put into the context of these motifs?

Thanks for the reviewer’s suggestion. We have rewritten this paragraph and the conserved amino acids are put into the context of the amino acid sequence motifs.

Line 201. The fluorescence changes associated with the E38A mutant appear slightly above background in empty liposomes.

We modified the sentence, and clarified the slight difference between mutants E38A and E106A.

Line 224-249 and Fig 3. The fluorescence changes are very low in these experiments. Were empty liposomes or proteoliposomes carrying an inactive mutant also used as a control to show that the changes are not due only to the leakage of protons down the electrical gradient?

In the revised manuscript, we included a negative control, which does not transport chlorhexidine in Supplementary figure S8. Together with Figure 2, we could prove that the fluorescence change observed in PA2880 preteoliposomes is indeed induced by PA2880, rather than the leakage of protons down the electrical gradient.

Line 225. From my understanding, chlorhexidine is divalent. The charged groups are also physically separated on opposite sides of the molecule and it is possible that two translocation cycles may occur during efflux via some pumps, eg. <https://doi.org/10.1038/ncomms5615>. Some discussion of these points could be included in the manuscript. This is very important as it also impacts the conclusion that that more than four H⁺ are exchanged for each substrate (lines 33, 88, 240).

In the revised manuscript, we deleted the description about the stoichiometry between chlorhexidine and protons. According to the literature, chlorhexidine has pKas of pKa1 = 7.63; pKa2 = 9.92; pKa3 = 8.22 and pKa4 = 10.52 (imine moieties). It should therefore carry several charges at pH 7. However, chlorhexidine might be protonated/deprotonated at different pH, and transporter PA2880 may select more neutral forms during the translocation cycle, making it difficult to precisely determine the coupling stoichiometry of PA2880. At this point, more evidence is needed to clarify this question.

Line 307. Rather than "The phylogenetic profiles of the two ...", I think the authors mean "The predicted orthologs of the two ..."

Following the reviewer's suggestion, we rephrased this sentence to "We identified orthologs of the two *A. baumannii* PACE family transporters, WP_002010078.1 (AceI) and WP_001161759.1 (AS_1503 in *A. baumannii* ATCC 17979) with fdog...".

Lines 413 and 417. The schematic in Figure 4 suggests that 10 uM chlorhexidine was used in these experiments, not 1 uM.

We corrected the chlorhexidine concentration to 10 μM, because actually 10 μM chlorhexidine was used in the experiment.

Supplement Table S2 "BW25113 (Δ acrB, Δ emfA, Δ emrE)" should be BW25113 (Δ acrB, Δ mdfA, Δ emrE).

In the revised manuscript, we corrected this mistake.

Supplement Figure S4. What is the molecular weight of PA2880? Are the authors confident of the monomer/dimer designations?

In the study of protein PA2880, a EGFP is fused to its C-terminal for easier detection. And our results showed that the presence of EGFP has no effect on the functional and assembly property of PA2880. Therefore, the protein molecular weight is 48.16 kDa. The bands shown in Supplementary Figure S4 corresponds well to protein molecular weight.

Supplement Figure S5. The purified PA2880 migrates at 150+ kDa relative to the markers in BN-PAGE. How do the authors interpret this with respect to the oligomeric state of the protein?

As described above, the molecular weight of protein PA2880-EGFP is 48.16 kDa. We could observe a major band in BN-PAGE at around 150 kDa, considering the effect of DDM, this should represent the dimeric protein.

Reviewer #2:

Zhao and coworkers present a study on the *Pseudomonas aeruginosa* protein PA2880, which belongs to the PACE family of transporters. Firstly, they present a phylogenetic analysis of the protein in order to determine its evolutionary relationship to other members of the family. The roles of conserved residues in the function of the protein are investigated by introducing mutations and comparing the function of the mutated proteins to the wildtype in minimum inhibitory concentration assays, transport experiments with protein reconstituted into liposomes and native mass spectrometry experiments. Understanding the function of PA2880 is a useful contribution to the field as little is known about the function of any proteins of the PACE family beyond the representative protein AceI. The protein of interest is also found in a pathogen, so unravelling its mechanism would be relevant for understanding and combating antimicrobial resistance. However, the data in this study is not strong enough to support the conclusions made and further investigations should be carried out to complete this work.

We appreciate reviewer's thoughtful and supportive comments. Our responses to your concerns are given below.

General remarks:

1. The authors conclude that PA2880 cannot transport cadaverine based on the MIC assays. However, in previously reported transport assays (in reference 14 of the manuscript) with reconstituted AceI, the cadaverine concentration used at pH 7.0 was 100 mM, which is far above the concentrations used in the MIC assays, so the observed effect could simply be a case of the cadaverine concentration used being too low rather than it not being a substrate.

In fact, up to 10 mg/ml cadaverine (97.8 mM) was used in the MIC assay. But both *E. coli* BL21 cells and *E. coli* cells expressing PA2880 cannot survive when cadaverine concentration is higher than 1.25 mg/ml, which means that the presence of PA2880 could not confer cells resistance to cadaverine when compared to control cells. Therefore, we made the conclusion that cadaverine is not the substrate of PA2880.

2. The data clearly shows that the W74A mutation perturbs the SDS-PAGE and SEC profiles of the protein as compared to the wild-type and other mutants. This raises concerns about the integrity of the W74A mutant protein, so this mutant should not be discussed further and neither should conclusions be made about this amino acid substitution from the experimental data.

Thanks for pointing out this problem. As reviewer suggested, the structure integrity of mutant W74A is indeed disrupted. Therefore, we deleted the description and discussion related to it.

3. Supplemental data 1 and 2 should contain the sequence alignments but they do not display as intended - this needs to be addressed.

Thank you very much for spotting this. We provide now the gene order information in the appropriate format.

4. In reference 14, which is the paper the current liposome transport assays are based on, it is noted that chlorhexidine may damage naked proteoliposomes. This puts the findings from the liposome assays under question. These conclusions need to be justified by performing controls (e.g. reconstituting a protein which does not transport chlorhexidine), statistics on the data to determine the significance of the difference between wild-type and mutant proteins or performing an independent experiment to corroborate the findings (e.g. with radiolabelled chlorhexidine)

As reviewer suggested, we selected a protein TqsA (An autoinducer 2 transporter, which does not transport chlorhexidine) as negative control to perform the transport assay in the same manner as we did for PA2880. As shown in Supplementary figure S8, protein TqsA behaves similar to the empty liposomes. Together with Figure 2, our results proved that PA2880 could mediate the transportation of chlorhexidine in the reconstituted system.

5. I would like to see the authors' working in more detail in relation to the calculation of more than 4 protons being exchanged for every molecule of chlorhexidine.

Following from the above point, I am unsure that quantitative observations could be made on the basis of the data from the liposome assays. Furthermore, chlorhexidine may be getting (de)protonated as it moves between environments at different pH, making a calculation which does not take this into account unreliable. The idea that four protons are transported, but nonetheless transport is primarily dependent on charge, not pH, seems surprising and should be supported by reference to other, similar, examples.

In the revised manuscript, we deleted the description about the stoichiometry between chlorhexidine and proton. According to the literature, chlorhexidine has pKas of pKa1 = 7.63; pKa2 = 9.92; pKa3 = 8.22 and pKa4 = 10.52 (imine moieties). It should therefore carry several charges at pH 7. However, chlorhexidine might be protonated/deprotonated at different pH, and transporter PA2880 may select more neutral forms during the translocation cycle, making it difficult to precisely determine the coupling stoichiometry of PA2880.

On the other hand, the electrogenic secondary active transporters could transport their substrates only in the presence of potential in the reconstituted system, for example, VcINDY and vSGLT (Fitzgerald et al, *Elife*, 2017), NCX_Mj (Shlosman et al, *J Gen Physiol*, 2018), EmrE (Kermani et al, *PNAS*, 2018) *et al.* And this property could be used to determine coupling stoichiometries of electrogenic secondary transporters reconstituted in proteoliposomes by measuring transporter equilibrium potentials (Fitzgerald et al, *Elife*, 2017).

6. The authors state that LILBID-MS, which they used, is comparable to nESI-MS, which is what Bolla et al (2020; cited as 'Robinson et al' in the manuscript) used to study AceI. However, there is no direct comparison made between the two studies. The authors should run a LILBID-MS experiment with AceI as Bolla and colleagues did to directly compare their findings. This would alleviate doubts about the effects observed being due to a discrepancy between the two methods, especially as the oligomeric states that Zhao and colleagues report are very different from those published about AceI - and yet the proteins are nonetheless clearly part of the same structural family. An effort to reconcile the Bolla et al. observation that the PACE proteins dimerise as a function of pH with the obligate dimer found here would be important.

As the reviewer suggested we ran an experiment with the AceI used by Bolla et al, however we could not reproduce the findings of their study.

As visible in the LILBID-MS spectra of AceI under the same sample conditions at pH 8, we could only detect monomeric AceI. This discrepancy does need further investigation, but would be beyond the scope of this study. It does not, however, contradict our findings for PA2880 as whatever reason is behind the lack of dimer for AceI cannot be influencing PA2880, as we detect a majority of dimer for this sample.

As shown in different earlier studies, LILBID-MS is reliable in determining the oligomeric state of proteins and detecting changes to it under changing sample conditions including pH value (Viet et al. Structure. 2019.; Peetz et al. J. Am. Soc. Mass Spectrom. 2018). In combination with the additional experiment in which the increase of the dimer population after addition of chlorhexidine was correctly shown by LILBID-MS, we stand with the results of our experiments.

From a mechanistical Standpoint, LILBID is better suited to reflect changes in pH compared to nESI. In nESI evaporation of solvent molecules leads to a concentration effect not only of the sample, but also concentrating the protons present in solution onto the sample molecule effectively changing the pH of the solution for the last milliseconds before the detection of the sample. Also, denaturation of parts of the sample due to the chain-ejection mechanism can alter the outcome of the measurement. None of this can happen in LILBID, where the sample is released from solution via a nanosecond laser pulse without any preceding charging of the sample solution with a high voltage.

7. On lines 32-33 in the Abstract the authors state that the dimerization of PA2880 is essential for its function (based on the mass spectrometry experiments). However, these experiments were only performed on the protein on its own. An experiment with the substrate chlorhexidine would shed more light on what the relationship between the substrate and the oligomerisation state is. No direct comparison can be drawn between the mass spectrometry data and the liposome assays since the protein is solubilised in DDM for LILBID-MS.

Following the reviewer's suggestion, we performed the LILBID-MS measurement in the presence of the substrate chlorhexidine. Using a substrate/protein molar ratio of 4, our results showed that chlorhexidine is bound to PA2880, and a significant increase in the dimer population was observed when chlorhexidine was added to the protein. To explore this further, we also performed the measurement using the substrate/protein molar ratio of 8. As shown in Figure 4d, the addition of chlorhexidine increased dimer formation. Taken together, the results proved that PA2880 should function as a dimer, and that the assembly of the dimer is supported by binding of chlorhexidine.

8. The English language in the manuscript is poor throughout (e.g. in the 'Importance' statement the first sentence on lines 39-40 refers to a pathogen being related to infections, which is the wrong word choice).

We revised the overall manuscript, and do hope that the language has been sufficiently improved

March 12, 2022

Prof. Hartmut Michel
Max Planck Institute of Biophysics
Molecular Membrane Biology
Max-von-Laue-Str. 3
Frankfurt am Main D-60438
Germany

Re: Spectrum01453-21R1 (Assembly and functional role of the PACE transporter PA2880 from *Pseudomonas aeruginosa*)

Dear Prof. Hartmut Michel:

Thank you for submitting the revised version of your manuscript. It has been accepted, and I am forwarding it to the ASM Journals Department for publication. You will be notified when your proofs are ready to be viewed.

Sincerely,

Ayush Kumar
Editor, Microbiology Spectrum
